# Role of Metabolomics in Pathogenesis and Prompt Diagnosis of Gastric Cancer Metastasis—A Systematic Review

**DOI:** 10.3390/diagnostics13223401

**Published:** 2023-11-08

**Authors:** Ștefan Ursu, Andra Ciocan, Cristina-Paula Ursu, Claudia Diana Gherman, Răzvan Alexandru Ciocan, Rodica Sorina Pop, Zeno Spârchez, Florin Zaharie, Nadim Al Hajjar

**Affiliations:** 1Department of Surgery, “Iuliu Hațieganu” University of Medicine and Pharmacy, Croitorilor Street, No. 19–21, 400162 Cluj-Napoca, Romania; stefan.ursu20@gmail.com (Ș.U.); cristinapaulapop10@yahoo.com (C.-P.U.); florinzaharie@yahoo.com (F.Z.); nadim.alhajjar@umfcluj.ro (N.A.H.); 2“Prof. Dr. Octavian Fodor” Regional Institute of Gastroenterology and Hepatology, Croitorilor Street, No. 19–21, 400162 Cluj-Napoca, Romania; 3Department of Surgery-Practical Abilities, “Iuliu Hațieganu” University of Medicine and Pharmacy, Marinescu Street, No. 23, 400337 Cluj-Napoca, Romania; gherman.claudia@umfcluj.ro (C.D.G.); razvan.ciocan@umfcluj.ro (R.A.C.); 4Department of Community Medicine, “Iuliu Hațieganu” University of Medicine and Pharmacy, Avram Iancu Street, No. 31, 400347 Cluj-Napoca, Romania; drsorinapop@yahoo.com; 5Department of Internal Medicine, “Iuliu Hațieganu” University of Medicine and Pharmacy, Croitorilor Street, No. 19–21, 400162 Cluj-Napoca, Romania; zsparchez@gmail.com

**Keywords:** gastric cancer, metabolomics, lipidomics, metastasis

## Abstract

Introduction: Gastric cancer is the fourth most frequently diagnosed form of cancer and the third leading cause of cancer-related mortality worldwide. The aim of this review is to identify individual metabolic biomarkers and their association with accurate diagnostic values, which can predict gastric cancer metastasis. Materials and Methods: After searching the keywords, 83 articles were found over a period of 13 years. One was eliminated because it was not written in English, and two were published outside the selected period. Seven scientific papers were qualified for this investigation after eliminating duplicates, non-related articles, systematic reviews, and restricted access studies. Results: New metabolic biomarkers with predictive value for gastric cancer metastasis and for elucidating metabolic pathways of the metastatic process have been found. The pathogenic processes can be outlined as follows: pro-oxidant capacity, T-cell inactivation, cell cycle arrest, energy production and mitochondrial enzyme impairment, cell viability and pro-apoptotic effect, enhanced degradation of collagen extracellular matrix, migration, invasion, structural protein synthesis, and tumoral angiogenesis. Conclusion: Metabolic biomarkers have been recognized as independent risk factors in the molecular process of gastric cancer metastasis, with good diagnostic and prognostic value.

## 1. Introduction

Gastric cancer is ranked the fourth most frequently diagnosed type of cancer globally and continues to be the third primary cause of cancer-related mortality [1,2,3]. Over the course of recent decades, there has been significant progress in the investigation of specific mechanisms of Helicobacter pylori infection, with research focusing on factors such as inherited susceptibility and environmental influences [4,5,6]. The main approach regarding targeted oncological treatment involves surgical resection, accompanied by adjuvant chemotherapy, or recently introduced radiation plus chemotherapy. These additional treatments have been shown to enhance the overall survival rate [1,7,8]. Regrettably, this disease is classified as one of the most aggressive oncological ailments with an unfavorable prognosis. Patients frequently present to the hospital in an advanced or metastatic stage, resulting in a significant burden, both economically and in terms of quality of life [9]. The median overall survival rate for individuals with advanced or metastatic gastric cancer remains below one year [10]. Therefore, it is imperative to prioritize preventive measures, early identification techniques, and innovative therapeutic strategies.

Using an “omics” approach, multiple studies are conducted on the metabolites and lipids present in gastric cancer tissue, plasma, or urine, with the goal of elaborating a metabolomic profile of this disease. The main scope is to identify crucial components playing a specific role in gastric carcinogenesis and to discover novel targets that might be potentially utilized as new lines of oncological treatment. The emergence of metabolomics has led to major advancements in comprehending the relationship between metabolic checkpoints and cancer [11]. Otto Heinrich Warburg, in the 1920s, demonstrated a distinctive metabolic pattern observed in all tumors, where neoplastic cells exhibit heightened glucose consumption through glycolysis, even in the presence of enough oxygen. This phenomenon is commonly referred to as the Warburg effect [12].

Extensive research has demonstrated that metabolic reprogramming is one of the main characteristics of cancer [13]. Moreover, it is intricately associated with oncogenesis [14,15], as well as evasion of the immune system by the neoplastic cells [16,17]. However, joining traditional research with metabolomics, lipidomics, and genomics is expected to yield more profound insights in understanding the main carcinogenic pathways and discover novel metabolites, in order to provide a more efficient and targeted treatment [11].

The aim of this review is to identify individual metabolic biomarkers and their association with an accurate diagnosis of gastric cancer metastasis.

Research questions:Which is the metabolite with potential diagnostic value for gastric cancer metastasis?What role does every metabolite have in understanding the molecular pathway of metastatic gastric cancer?

## 2. Materials and Methods

The methodology of this systematic review consists of defining search algorithms, selection criteria, and data extraction protocols. The present research followed the guidelines outlined in the Preferred Reporting Items for Systematic Reviews and Meta-Analyses (PRISMA) statement (Figure 1). From January 2010 until September 2023, articles published in the English language in the online databases PubMed (Medline), Embase, and Clarivate Web of Science were analyzed.

The research was performed using the following keywords: “gastric cancer” OR “gastric carcinoma” AND “metabolomics” AND “metastasis” using “AND”, “OR” between the mentioned elements as Boolean Operators. All titles referred to in English and published in a precise period of 13 years were assessed for eligibility by title and abstract by two separate researchers to remove duplications.

Inclusion criteria: all studies including information about the use of metabolomics in the diagnosis of gastric cancer metastasis; type of study: original article, clinical trial, and randomized control trial; type of subjects: patients, cell cultures, or animals.

Exclusion criteria: articles published earlier than 2010, not referring to the subject, letters to the editor, short reports, meta-analyses, systematic reviews, narrative reviews, non-English papers, patients under 18 years old, or articles focusing on proteomics, genomics, and primary gastric malignant tumor diagnosis. Furthermore, studies regarding the application of metabolomics and lipidomics in benign gastrointestinal disorders and acute inflammatory processes were also excluded.

In order to evaluate the risk of bias in every individual study, we applied the updated quality assessment of diagnostic accuracy studies-2 (QUADAS-2) method. This approach covers four areas of bias, namely, patient selection, index test, reference standard, and patient flow and timing. An Excel extraction tool was utilized for the purpose of data collection. For the quality of the study evaluation, we utilized criteria including study participation, factor measurement, value, and applicability. Following a thorough evaluation of all eligible articles, two evaluators systematically collected data and cross-verified all findings. In the process of data selection and extraction, any inconsistencies identified by the two primary reviewers were subjected to further examination by two additional reviewers. The list of references of specific research projects was systematically examined to identify potential publications through the application of the snowball technique.

During a timeframe of 13 years, after searching the combination of keywords mentioned, a number of 83 articles, containing data on metabolite identification for metastatic gastric cancer, were found. From this selection of articles, one was eliminated because it was not written in English and two of them were published outside the selection period. All duplicated articles, publications not referring to the matter, systematic reviews, and access restriction papers were eliminated, thus making seven scientific papers eligible for this study.

## 3. Results

In this present study, all the articles included (*n* = 7) focused on identifying new potential metabolomic biomarkers for gastric cancer metastasis and elucidating the pathogenic and metabolic pathways of the metastatic process (Table 1).

Among the evaluated articles, five of them used human subjects and the rest conducted studies on an animal model: male mice with severe combined immune deficiency (SCID) with human gastric cancer SCG-7901 cell line [18,19]. Five of the studies used tissue samples for the metabolomic analysis, two used plasma [20,21], one used urine [18], and one used peritoneal fluid [22] (Table 2).

**Table 1 diagnostics-13-03401-t001:** Author name, year of publication, title of the article, and the aim of the study.

Author Name	Year of Publication	Title of the Article	Aim of the Study
Hu et al. [18]	2011	Prediction of gastric cancer metastasis through urinary metabolomic investigation using * GC/MS	Identifying metabolomic biomarkers of gastric cancer invasiveness and elucidating the underlying mechanisms of metastasis
Chen et al. [19]	2010	Metabolomics of gastric cancer metastasis detected by gas chromatography and mass spectrometry	Identifying metabolomic biomarkers of gastric cancer invasiveness and elucidating the underlying mechanisms of metastasis
Shi et al. [20]	2021	Abnormal arginine metabolism is associated with prognosis in patients with gastric cancer	Investigating the function of arginine in pathogenesis and its prognostic significance in metastatic gastric cancer
Pan et al. [22]	2020	Discovering biomarkers in peritoneal metastasis of gastric cancer by metabolomics	Evaluating the role of metabolomics in gastric cancer peritoneal metastases
Gu et al. [21]	2015	Perioperative dynamics and significance of amino acid profiles in patients with cancer	Evaluating the role of metabolomics in gastric cancer peritoneal metastases
Zhang et al. [23]	2018	* H NMR metabolic profiling of gastric cancer patients with lymph node metastasis	Identifying metabolomic biomarkers for carcinogenesis, invasion, and metastasis in gastric cancer. The first metabolomic study of lymph node metastasis in gastric cancer
Sun et al. [1]	2020	Activation of * SREBP-1c alters lipogenesis and promotes tumor growth and metastasis in gastric cancer	Identifying metabolomic biomarkers in gastric cancer tissue, relevant lipids, and primary upstream regulatory factors

* GC-MS, gas chromatography–mass spectrometry; ^1^H NMR, proton nuclear magnetic resonance; SREBP-1c, sterol regulatory element-binding protein-1.

**Table 2 diagnostics-13-03401-t002:** Author name, year of publication, subject type, biological product used, and the levels of metabolites (increased/decreased) mentioned in every article.

Author Name, Year of Publication	Patients or Animal Subjects/Sample Size	Biological Product (Method Used)	Increased Levels of Metabolites in Biological Samples	Decreased Level of Metabolites in Biological Samples
Hu et al., 2011 [18]	Male SCID mice Human gastric cancer * SCG-7901 cell line (intestinal-type adenocarcinoma) Gastric cancer group (*n* = 16) Metastatic (=8) Non-metastatic (=8) Control group (*n* = 8)	Urine (* GC-MS)	Myo-inositol Butanedioic acid	L-proline Alanine Glycerol Butanoic acid L-threonic acid
Chen et al., 2010 [19]	Male SCID mice Human gastric cancer SCG-7901 cell line (intestinal-type adenocarcinoma) Gastric cancer group (*n* = 16) Metastasis (=8) Non-metastasis (=8) Control group (*n* = 6)	Tissue sample (GC-MS)	Myo-inositol Lactic acid L-alanine L-valine Leucine Malic acid L-aspartic acid Serine Proline Phosphoserine Dimethylglycine Glycine L-glutamic acid L-lysine Propanedioic acid Docosanoic acid Octadecanoic acid Arginine Pyrrodine Pyrimidine	Butanedioic acid L-threonic acid Glucose Succinate L-isoleucine L-methionine Propanamide Glutamine Hypoxanthine
Shi et al., 2021 [20]	Human subjects Total subjects (*n* = 454) Gastric cancer group (=92) (intestinal adenocarcinoma, mixed and diffuse type) Gastric ulcer group (= 51) Gastric polyps group (=206) Gastritis group (=105)	Plasma (* LC-MS/MS)	-	Arginine
Pan et al., 2020 [22]	Patients Total subjects (*n* = 62) (histological type not mentioned)	Peritoneal lavage fluid (LC-MS)	Sulfite G3P Cl (63:4) * PE-NMe TG (54:2) α-aminobutyric acid α-CEHC Dodecanol Glutamyl alanine 3-methylpropionic acid Retinol 3-hydroxysterol Tetradecanoic acid * [MG (21:0/0:0/0:0)] Tridecanoic acid Myristate glycine Octadecanoic acid * TG (53:4)	-
Gu et al., 2015 [21]	Patients Gastric cancer group (*n* = 56) (intestinal-type adenocarcinoma) Breast cancer group (*n* = 28) Thyroid cancer group (*n* = 33) Healthy age-matched control group (*n* = 137)	Plasma (LC-MS/MS)	-	Threonine Histidine * EAAs * GAAs
Zhang et al., 2018 [23]	Patients Total subjects (*n* = 120) (intestinal-type adenocarcinoma and diffuse-type signet ring cells) LNM-positive GC group (=40) LNM-negative GC group (=40) Normal control group (=40)	Tissue sample (* H NMR spectroscopy)	Isoleucine Leucine Valine Glutathione	Glycine Choline Betaine Tyrosine Hypoxanthine
Sun et al., 2020 [1]	Patients AGS cells SGC-7901 cells MGC-803 cells GES-1 cells Gastric cancer group (*n* = 29) (intestinal-type adenocarcinoma) Control group (*n* = 20)	Tissue sample (* UPLC-MS/MS)	-	Palmitic acid

* α-CEHC, α-carboxy-ethyl-hydroxychromanol; EAAs, essential amino acids; GAAs, glucogenic amino acids; G3P, glyceraldehyde-3-phosphate; LC-MS/MS, liquid chromatography–tandem mass spectrometry; LNM, lymph node metastasis; [MG (21:0/0:0/0:0)], monoradyglycerols; PE-NMe, monomethylphosphatidylethanolamine; TG, triglyceride; UPLC-MS/MS, ultra-performance liquid chromatography.

The analysis of tissue samples revealed that the concentrations of various metabolites were significantly reduced in individuals with metastatic disease compared to the healthy control group. Specifically, glycine, choline, betaine, tyrosine, hypoxanthine, palmitic acid, glucose, succinate, L-isoleucine, L-methionine, propanamide, glutamine, L-threonic acid, and butanedioic acid exhibited lower levels in the metastatic disease group [1,19,23].

The urinary metabolomic profile of metastatic gastric cancer was investigated by Hu et al. The study utilized male SCID mice that were inoculated with the human gastric cancer SCG-7901 cell line. The findings revealed that certain metabolites, such as L-proline, alanine, glycerol, butanoic acid, and L-threonic acid, showed lower baseline levels in the metastatic group. On the other hand, increased levels of myo-inositol and butanedioic acid were observed in comparison to the non-metastatic group [18].

In their study, Chen et al. used male SCID mice that were incorporated with the human gastric cancer cell line SCG-7901. The researchers aimed to identify the metabolomic profile signature of metastatic gastric cancer by analyzing tumoral tissue [19]. Through their investigation, they observed that three significant metabolic biomarkers (proline, serine, and arginine) exhibited elevated baseline levels in the analyzed samples.

A study investigating metabolites in peritoneal lavage fluid among a cohort of 62 patients diagnosed with gastric cancer was performed by Pan et al. [22].

Each one of the publications considered in this review highlighted several significant discoveries (Table 3). Besides showing differences regarding metabolites in the metastatic group versus the non-metastatic group, the randomized control trial conducted by Hu et al. underlined the diagnostic value of both lactic and butanoic acids, underlying seven metabolites involved in gastric cancer metastatic model [18].

Chen et al. pointed out that in the metastatic group, 20 metabolites exhibited up-regulation, whereas 9 metabolites displayed down-regulation, as indicated in the tumor models [19]. Arginine levels were notably increased in individuals diagnosed with non-metastatic gastric cancer in comparison to those presenting metastases as Shi et al. mentioned in their study. Moreover, arginine can be used as an independent prognostic factor of the oncological treatment. Its overexpression has been linked to prolonged patient survival [20].

Pan et al. identified 18 metabolites with considerable diagnostic potential in identifying peritoneal metastases of gastric cancer [22].

Regarding alanine, Gu et al. stated that this agent has a distinct role in the field of cancer biology by exerting inhibitory effects on the proliferation of gastric cancer cells, in conjunction with glutamine, while simultaneously boosting the proliferation of breast cancer cells. Along with these findings, the study reveals that threonine, histidine, essential amino acids, and glucogenic amino acids are significantly correlated with lymph node metastasis [21].

Additionally, Zhang et al. have demonstrated the presence of eight metabolites having noteworthy discriminatory capacity in separating subjects with lymph node metastases from those without [23].

Individuals diagnosed with gastric malignancy demonstrated a higher concentration of palmitic acid (PA) in their serum, compared to the control group, as previously shown by Sun et al. [1].

Upon conducting an analysis of the studies, a series of metabolites and their associated pathogenic processes have been identified (Table 4). The pathogenic processes can be outlined as follows: pro-oxidant capacity, T-cell inactivation, cell cycle arrest, energy production and mitochondrial enzyme impairment, cell viability and pro-apoptotic effect, enhanced degradation of collagen extracellular matrix, migration, invasion, structural protein synthesis, and tumoral angiogenesis. Furthermore, it should be noted that only certain metabolites are involved in certain aspects of the pathogenic process. Arginine expresses effects in the process of protein synthesis, tumoral angiogenesis, T-cell inactivation, malignant cell activity, and pro-apoptotic out-turns. Lactic acid acts as an important agent in the degradation of the extracellular matrix, neoplastic angiogenesis, and T-cell dysfunction and it is an energy generator. Additionally, one of the most encountered metabolites is L-proline, which has deteriorating effects on mitochondrial enzymes.

## 4. Discussion

Gastric cancer represents a significant hazard to human health due to its high incidence and unfavorable prognosis [2,24]. The majority of deaths associated with gastric cancer appear in the metastatic phase, even after curative gastrectomy [11]. The occurrence of distant organ metastasis, particularly peritoneal metastases, which represents the most typical form of disease recurrence, plays a significant role in mortality. The vast majority of patients diagnosed with metastatic gastric cancer have typically developed malignant ascites, forfeiting the chance of therapeutic intervention [25,26]. Nevertheless, the incipient stages of peritoneal carcinomatosis fail to develop any symptoms or signs, and traditional imaging techniques such as ultrasound and CT scans are unable to accurately diagnose infracentimetric peritoneal nodules. Regrettably, there are currently no available molecular markers, which may be utilized for the prediction of metastasis. Therefore, there is an urgent requirement to identify more sensitive diagnostic markers for metastatic gastric cancer. In contrast to certain individual molecular markers, metabolic markers exhibit more comprehensiveness and accuracy [27,28,29,30,31,32,33,34,35,36,37]. Metabolomics has emerged as a significant tool in the identification and evaluation of cancer biomarkers, garnering considerable attention in contemporary research. According to reports, the metabolic process of glucose has been identified as a significant factor in the progression of gastric cancer. However, a few clinical studies have revealed that certain lipid metabolites are significantly involved in the peritoneal metastases of gastric cancer [38]. This phenomenon can be attributed to a large variety of pathogenic mechanisms associated with gastric cancer.

Because it is a non-invasive priority, the investigation of gastric cancer biomarkers in blood or urine is increasingly valued. All studies showed that the metastatic groups had lower glucose levels and higher lactic acid levels, the byproduct of glycolysis, than the non-metastatic group. Tumoral cells increase the production of lactic acid, which leads to T-cell inactivation, acid-mediated matrix breakdown, vascular endothelial growth factor (VEGF), hypoxia-inducible factor (HIF-1) up-regulation, and increased cell motility, all of which provide a favorable environment for dissemination [39,40]. A frequently observed phenomenon in almost all cancer cells is the increased glucose uptake and fermentation to lactic acid, under normoxic circumstances of the environment [11,19].

Implementing animal models with gastric cancer cell line SGC-7901, Chen et al. used metabolomics to confirm that, with an AUC value of 1.0, biomarkers associated with proline and serine metabolism could discriminate metastatic from non-metastatic tissue [11,19]. The most highly up-regulated tissue metabolite was proline, suggesting a potential correlation between the accelerated turnover of extracellular matrix in metastatic cancer cells and elevated proline levels in gastric cancer tissue that has spread to other sites [41]. Due to pyrroline-5-carboxylic (P5C), it is connected to the pentose phosphate pathway (PPP), the tricarboxylic acid cycle, and the metabolism of arginine and glutamate [42]. This indicates that the considerably elevated metabolism of proline is strongly associated with cancer dissemination. Additionally, the metastatic specimens had lower levels of methionine and threonine and higher amounts of leucine, valine, glutamate, and lysine, suggesting that the need for energy increases with the advancement of metastatic disease.

In a study conducted by Hu et al., the investigators explored the relationship between metabolite levels and the distinction between non-metastatic and metastatic groups in male SCID mice with human gastric cancer SCG-7901 cell line. The findings of the study showed that reduced levels of alanine, glycerol, L-proline, butanoic acid, and L-threonic acid, accompanied by elevated levels of butanediotic acid and myo-inositol, were able to accurately identify the non-metastatic and metastatic groups. Extensive data regarding the metabolic composition of urine in both healthy individuals and those with cancer have been thoroughly examined. This investigation, in conjunction with the principal component analysis (PCA) model, has proven to be effective in identifying distinctive metabolic alterations associated with gastric cancer. Specifically, a set of seven metabolites has been carefully chosen to construct a diagnostic model capable of distinguishing between non-metastatic and metastatic gastric cancer cases [11,18].

Abnormal arginine metabolism was carefully investigated by Shi et al., who demonstrated that the plasma arginine index has been demonstrated to be substantially higher in individuals diagnosed with non-metastatic gastric cancer (stages I, II, and III) compared to those with metastatic gastric cancer (stage IV). Patients diagnosed with moderately differentiated (G2) gastric adenocarcinoma had elevated levels of plasma arginine compared to patients with poorly differentiated (G3) gastric cancer. Furthermore, there was a negative correlation observed between plasma arginine levels and tumor markers, both carcinoembryonic antigen (CEA) and carbohydrate antigen 19-9 (CA19-9) [20,43]. Although plasma levels of arginine are lower in individuals with cancer compared to healthy controls and the benign pathology group, the opposite trend is observed in tissue samples, where arginine levels are significantly raised [19].

The presence of oleic acid in omental adipocytes has been proven to exacerbate the invasiveness of gastric cancer cells, which has been generated by the activation of the protein kinase B (PI3K-Akt) signaling pathway [44]. In their study, Sun et al. conducted an analysis of the lipidomic profile in individuals with malignant gastric tumors, aiming to investigate the potential variations in palmitic acid levels between cancer cases and control subjects. The findings of their research indicated that there was no statistically significant distinction observed in the levels of PA between the two groups. In comparison to the control group, individuals diagnosed with gastric cancer showed a slightly elevated level of PA in their serum; however, the observed disparity did not reach statistical significance [1]. Furthermore, it has been noted that a decreased concentration of PA (with downregulation of the SCD1 gene), specifically at 100 μM, facilitated the migration and invasion of gastric cancer cells (MGC-803 and HGC27) by activating AKT [45].

Concerning the most common site of metastasis (peritoneal dissemination) and the significant role of lipid metabolism in pathogenic mechanisms of gastric cancer metastasis, by analyzing peritoneal lavage fluid, Pan et al. [22] aimed to identify new non-invasive biomarkers with high sensitivity and specificity. A panel of metabolites that exhibited promising diagnostic capabilities was successfully identified: TG (54:2) refers to a specific type of triglyceride molecule with a fatty acid composition of 54 carbons and 2 double bonds. G3P stands for glycerol-3-phosphate, which is an important intermediate in various metabolic pathways. Alpha-aminobutyric acid is an amino acid derivative that plays a role in neurotransmission. Alpha-CEHC is an abbreviation for α-tocopherol quinone, a metabolite of vitamin E. Dodecanol is a 12-carbon alcohol compound. Glutamyl alanine is a dipeptide composed of the amino acids, glutamic acid and alanine. The metabolite 3-methylalanine is an amino acid with a methyl group attached to the carbon chain. Sulfite is a chemical compound containing sulphur and oxygen. CL (63:4) refers to a specific type of cardiolipin molecule with a fatty acid composition of 63 carbons and 4 double bonds. PE-NMe (40:5) is a phosphatidylethanolamine derivative with a fatty acid composition of 40 carbons and 5 double bonds. TG (53:4) is another type of triglyceride molecule with a fatty acid composition of 53 carbons and 4 double bonds. Additionally, these metabolites were found to potentially serve as independent risk factors for gastric cancer metastasis [22,46].

Cancer patients displayed substantially higher concentrations of threonine, arginine, and essential amino acids, whereas they had significantly lower plasma concentrations of aspartic acid, glucose, glycine, proline, and non-essential amino acids (NEAAs) due to increased purine and pyrimidine synthesis to sustain tumoral growth, as mentioned by Gu et al. in their study [21]. Under certified guidance, the role of essential and non-essential amino acids involved in the pathogenetic process of gastric cancer metastasis was evaluated, and the levels of threonine, histidine, essential amino acids, and glucogenic amino acids were significantly related to lymph node metastasis in patients with malignant gastric tumors.

A major prognostic factor in gastric cancer, especially for the ones with early gastric cancer (EGC), is the presence of lymph node metastasis. The majority of subjects with LNM-positive gastric cancer experienced a significantly lower overall survival rate in comparison to the LNM-negative [47,48]. There has been an increasing amount of studies suggesting a disruption in the metabolism of branched-chain amino acids. In their study, Zhang et al. successfully identified a total of 33 metabolites serving as distinguishing factors between gastric cancer tissues and normal control samples. These metabolites include glutamine, acetic acid, alanine, threonine, citrulline, lactate, valine, leucine, isoleucine, N-acetyl glycoprotein, O-acetyl glycoprotein, VLDL, glucose, myo-inositol, acetone, D-ribose, Lipid-CH2-C=O, succinate, methylamine, phosphocholine, taurine, trimethylamine-N-oxide, pyruvate, glutathione, choline, lysine, betaine, glycine, tyrosine, serine, uracil fumarate, and hypoxanthine. A set of eight metabolites, namely, branched-chain amino acids including leucine, isoleucine, valine, as well as glycine, glutathione, betaine, hypoxanthine, and tyrosine, have proven significant discriminatory capability in distinguishing LNM-positive subjects from the LNM-negative ones. Moreover, they hold promise as potential factors for diagnosis and prognosis in gastric cancer [23].

An important aspect in the accurate metabolic profile evaluation of the disease is the deep analysis before any treatment such as chemotherapy or radiation, because these cause a severe alteration in the metabolism. The majority of articles included in our study (*n* = 5) have the neoadjuvant oncological treatment listed as an exclusion criterion [1,18,19,22,23]. Shi et al. and Gu et al. eliminated this aspect at the expense of a larger number of patients in the cohort evaluated [20,21]. Moreover, owing to the fact that each histological type has different behavior in the evolution of the disease, metastasis process, and overall survival, a subgroup analysis was carried out in order to emphasize the metabolic variations. Only two articles subdivided the included cases using the Lauren classification [20,23]. Comparing the arginine levels in 92 gastric cancer patients, Shi et al. found no significant statistical correlation between the intestinal type and the diffuse type, in contrast with Zhang et al., who only used the histological types for the clinical characterization of the included patients and not for metabolites analysis based on histological features.

The limitations of our systematic review are mostly attributed to the absence of standardized metabolites used for this particular condition, as well as the inclusion of diverse techniques for analysis within our investigation. The majority of the studies included were original articles, where the research was conducted on small numbers of patients, and two of them included gastric cancer cell cultures inoculated in male SCID mice [18,19], here emerging the main source of bias. The scarce number of publications available in the literature suggests a lack of comprehensive understanding on the subject matter. Hence, the application of excessively stringent exclusion criteria is impossible due to the data being diverse and based on the number of cases undergoing metabolomic analysis in each article, the histological type of gastric cancer, and neoadjuvant oncological treatment such as chemotherapy and radiation received by the patient prior to evaluation; barriers that are contingent upon the researcher’s preferences or, more importantly, upon the available resources.

To determine the effectiveness, specificity, and sensibility in detecting early gastric cancer metastasis and possible benefits, if any, that these series of metabolites may offer, randomized studies contrasting metabolomic analysis in patients receiving neoadjuvant chemotherapy/radiation and without oncological treatment are required. Moreover, studies regarding each histological type of gastric cancer, such as intestinal type (adenocarcinoma) and diffuse type (signet ring cell), in order to find a metabolic signature for early diagnosis are mandatory due to the fact that the invasiveness and behavior are different based on the histology of the disease.

Metabolomics is situated down the road from proteomics, genomics, and transcriptomics, as it aims to comprehensively characterize the alterations in the metabolic processes occurring in response to certain situations, including the presence of pathogenic factors, host factors, or environmental co-effectors [11]. Nonetheless, it is imperative to integrate metabolomics with other-omics methodologies in order to achieve a greater understanding of gastric carcinogenesis. The relationship between the microbiome and the metabolome has garnered significant attention due to emerging evidence that disruptions in the structure or function of gastrointestinal bacteria, such as Helicobacter pylori, can contribute to the development of gastric cancer [49,50]. Therefore, it is justifiable to hypothesize that the examination of gastric flora might be integrated into a comprehensive investigation of the prevalent metabolic issues related to gastric cancer. However, it is important to note that additional research is needed to bridge the current gap in research.

## 5. Conclusions

Gastric cancer is widely recognized as an aggressive malignancy with significant global implications for public health. Despite the lack of understanding of its pathophysiology, important advancements have been made in the field of -omics investigations, offering potential insights. Metabolomics provides a comprehensive understanding of metabolic disturbances in the stomach, distinguishing between healthy and malignant conditions and facilitating the identification of disease-specific biomarkers.

Metabolic biomarkers mentioned in this study have good diagnostic and prognostic value; some of them are even considered as independent risk or predictive factors, with an important role in the pathogenesis of gastric cancer metastasis, such as energy production, structural protein synthesis, pro-apoptotic effect, cell cycle arrest, antioxidant effect, tumor angiogenesis, mitochondrial enzyme impairment, T-cell dysfunction, and other metabolic pathways involved in cell viability, migration, and invasion.

With the continuous progress of technology and the increasing knowledge regarding metabolic disturbance in gastric cancer, it is expected that novel diagnostic and therapeutic targets will inevitably arise. In conclusion, these advancements have the potential to be implemented in clinical settings, thereby achieving the objective of genuinely individualized cancer therapy.

## Figures and Tables

**Figure 1 diagnostics-13-03401-f001:**
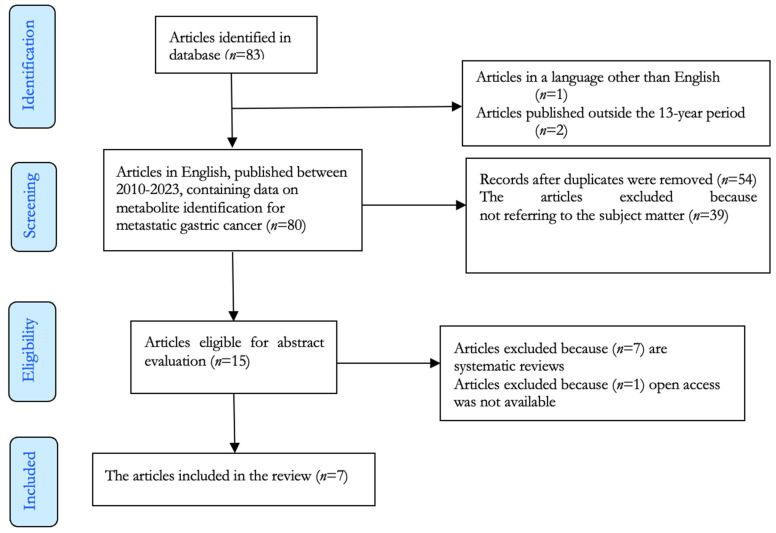
PRISMA flow diagram for the selected studies included in the systematic review (records identified from PubMed, Embase, and Web of Science databases).

**Table 3 diagnostics-13-03401-t003:** Author name, year of publication, and major significant findings of the articles included.

Author Name	Year of Publication	Significant Findings
Hu et al. [18]	2011	Differences with regard to 10 metabolites between the gastric cancer group (metastatic and non-metastatic groups) and the control group—lactic acid, malic acid, butanoic acid, citric acid, glycerol, hexadecanoic acid, pyrimidine, uric acid, propanoic acid, and butanedioic acid. There were 7 distinct metabolites that exhibited distinctive differences between metastatic cancer and non-metastatic cancer—alanine (Ala), L-proline (Pro), glycerol, butanoic acid, butanedioic acid, L-threonic acid, and myo-inositol. The diagnostic value of changes in lactic acid and butanoic acid has been demonstrated. Gastric cancer metastatic model has been constructed by a sequence of 7 metabolite markers.
Chen et al. [19]	2010	29 distinctive metabolites had different levels of expression between the metastatic and non-metastatic group—lactate (Lac), alanine (Ala), propanedioic, leucine (Leu), glycine (Gly), proline (Pro), serine (Ser), valine (Val), pyrimidine, dimethylglycine, succinate, isoleucine, propanamide, butanedioic, pyrrolidine, malic acid, methionine, threonine (Thr), glucose, glutamine (Glu), aspartic (Asp), phosphoserine, glutamate (Glu), lysine (Lys), hypoxanthine, arginine (Arg), insositol, octadecaoic, and docosanoic. 20 metabolites mentioned in the tumor models were up-regulated and 9 metabolites were down-regulated in the metastasis group. Proline and serine metabolisms are involved in the metastasis process of gastric cancer.
Shi et al. [20]	2021	The plasma concentrations of arginine were shown to be significantly elevated in individuals diagnosed with non-metastatic gastric cancer (stages I, II, and III) compared to those with metastatic gastric cancer. Arginine level before oncological treatment can be used as an independent prognostic factor. High arginine overexpression has been associated with long-term survival of the patient. The upregulation of argininosuccinate synthase 1 (ASS1) was associated with a significant extension in the overall survival of individuals diagnosed with gastric cancer.
Pan et al. [22]	2020	The following 18 metabolites: TG (54:2), G3P, α-aminobutyric acid, α-CEHC, dodecanol, glutamyl alanine, 3-methylalanine, sulfite, CL (63:4), PE-NMe (40:5), TG (53:4), retinol, 3-hydroxysterol, tetradecanoic acid, MG (21:0/0:0/0:0), tridecanoic acid, myristoyl glycine, and octacosanoic acid possess significant diagnostic potential for peritoneal metastasis in gastric cancer. Fatty acids have the potential to serve as an early detection marker and independent predictive factor for gastric cancer.
Gu et al. [21]	2015	Cancer patients displayed notably elevated concentrations of Thr, Arg, and essential amino acids (EAAs), while experiencing considerably reduced levels of Asp, Glu, Gly, Pro, non-essential amino acids (NEAAs), and ammonia (NH3), in comparison to healthy controls. Patients diagnosed with gastric cancer exhibited dramatically reduced levels of serine, alanine, valine, lysine, histidine, branched-chain amino acids (BCAAs), glucogenic amino acids (GAAs), and total amino acids (TAAs). Lymph node metastases were correlated with increased levels of threonine (Thr), histidine (His), essential amino acids, and glucogenic amino acids. Notable elevation in the concentrations of methionine, leucine, tyrosine, and lysine is seen in individuals diagnosed with thyroid cancer. Alanine is playing a specific role in cancer biology by expressing inhibitory effects on the proliferation of gastric cancer cells, in comparison to glutamine, which promotes cell proliferation in breast cancer.
Zhang et al. [23]	2018	A total of 33 metabolites were successfully identified as differentiating factors between gastric cancer tissues and normal control samples: glutamine, acetic acid, alanine, threonine, citrulline, N-acetyl glycoprotein, O-acetyl glycoprotein, lactate, valine, leucine, isoleucine, very-low-density lipoprotein (VLDL), glucose, myo-inositol, acetone, D-ribose, Lipid-CH2-C=O, succinate, pyruvate, glutathione, choline, methylamine, phosphocholine, taurine, trimethylamine-N-oxide, lysine, betaine, glycine, serine, uracil, tyrosine, fumarate, and hypoxanthine. A total of 8 metabolites have shown significant discriminatory ability in distinguishing between lymph node metastasis (LNM)-positive and LNM-negative individuals with gastric cancer: branched-chain amino acids (BCAAs: leucine, isoleucine, valine), glycine, glutathione, betaine, hypoxanthine, and tyrosine.
Sun et al. [1]	2020	The activation of SREBP-1c led to alterations in lipogenic enzymes, including increased expression of stearoyl-CoA desaturase 1 (SCD1) and fatty acid synthase (FASN) and decreased expression of fatty acid elongase 6 (ELOVL6). The combined effect of these enzymes has caused a decrease in the concentration of palmitic acid. In comparison to the control group, the gastric cancer group exhibited an elevated serum concentration of palmitic acid.

**Table 4 diagnostics-13-03401-t004:** Main pathogenic processes and the involved metabolites.

Energy Production	Structural Protein Synthesis	Pro-Apoptotic Effect Cell Cycle Arrest	Antioxidant Capacity	Tumoral Angiogenesis
Glucose Lactic acid Alanine Glycerol TG (54:2) PE-NMe Cl (63:4) TG (53:4) MG (21:0/0:0/0:0:0:0) Myristate glycine Tridecanoic acid Octadecanoic acid 3-methylpropionic acid Tetradecanoic acid Dodecanol Succinate Malic acid Serine Glyceraldehyde-3-phosphate	L-proline Arginine Glycine Serine Aspartic acid Glutamic acid Glutamine Valine Methionine Histidine Isoleucine Leucine Lysine Phenylalanine Threonine Tryptophan	L-proline Arginine Sulfite Cystein Methionine Alanine Glutamine	Glutathione Choline Betain Homocysteine	Lactic acid Arginine Glyceraldehyde-3-phosphate
**Enhanced Degradation of Collagen Extracellular Matrix**	**Mitochondrial Enzyme Impairment**	**T-Cell Dysfunction/Inactivation**		**Cell Viability, Migration, and Invasion**
Lactic acid L-proline	Tricarboxylic acid (TCA) intermediates Butanedioic acid Malic acid Citric acid L-proline	Lactic acid Arginine		Palmitic acid Sulfite Retinol Arginine

## Data Availability

Data are contained within the article.

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
