# Peer review of "Role of Metabolomics in Pathogenesis and Prompt Diagnosis of Gastric Cancer Metastasis—A Systematic Review"

_diagnostics, 2023, doi:10.3390/diagnostics13223401_

Round 1

Reviewer 1 Report

Comments and Suggestions for Authors

In this manuscript, the authors systematically reviewed the role of metabolomic biomarkers in predicting the occurrence of metastasis as well as the pathogenesis of metastasis in the course of gastric carcinoma. To me, the idea is interesting, the study is well-conducted, and the manuscript is well-written.  

Major points:

1. Gastric carcinomas are generally classified into two pathophysiologically and clinically relevant histologic types: intestinal-type and diffuse-type (signet-ring cell). Did the authors perform any subgroup analyses for the type of gastric carcinoma? Please include the type of gastric carcinoma investigated in each study in Table 2.

2. A potential confounder would be the type of therapy given to the patients in each study. Were there any significant differences in the treatment modalities between the studies?

3. Lines 81-82: I think AND is not a suitable Boolean operator between “gastric cancer” and “gastric carcinoma”. I guess OR has been used instead.

4. Lines 179-182: These lines are best attributed to the study by Shi et al. (reference 20), but they have been linked to Chen et al. (reference 19).

Minor points:

1. There are typos. For example: “diagnostic” in the title (suggested word: diagnosis); “World of Science” in Figure 1’s legend;

2. Line 89: I think 2013 should be replaced by 2010.

3. Line 140: Please make a citation for the study conducted on urine.

4. Table 2, footnote: It is recommended to sort abbreviations alphabetically.

5. Line 178: It is written that 9 metabolites were downregulated, while 8 metabolites are mentioned in Table 1.

Author Response

Thank you for the useful comments. We have revised our manuscript and answered all the requirements:

Major points

  1. We have added the histological type of gastric carcinoma in every study evaluated, both in Table 2 and Discussion section.
  2. The neoadjuvant systemic therapy is an important aspect of difference in metabolites' assessment between studies, 5 of them took this into consideration, the other two could not afford to lose patients because of this aspect, since the number of subjects is already limited.
  3. We have corrected this in the Material and methods section, as you correctly suggested.
  4. The phrase was too long and the reference at the end was misleading, we have reorganised the paragraph for a more cursive presentation of information.

Minor points

  1. Thank you for your fine observations, we have reanalysed the entire manuscript for language, grammar and typing inaccuracies.
  2. Yes, indeed. We have modified.
  3. We have added reference 18 to the study conducted on urine analysis - now line 151.
  4. We have now arranged the abbreviations in an alphabetical order.
  5. We have omitted hypoxanthine from the Table 1, but now we added it. There are 9 metabolites (it was correct in the text), we omitted the last one in the table.

Additional information has been provided in the Materials and Methods section and also in the Discussion section regarding all the selection process, study limitations and also in between studies comparison.

Text english has been revised and modified in order to provide a more clear logical relationship and a better understanding of the manuscript. 

Reviewer 2 Report

Comments and Suggestions for Authors

This article reviews the potential applications and significance of metabolic biomarkers in the diagnosis of gastric cancer metastasis in the past decade from the perspective of metabolomics. The review focuses on the discovery of new metabolic biomarkers for gastric cancer metastasis and elucidates their mechanisms of action and metabolic pathways. This is a meaningful work that provides potential methods and ideas for the diagnosis and treatment of gastric cancer. However, there are still some questions, which are listed as follows:

1)Are the seven articles mentioned in the manuscript with analytical value too few in quantity to support the research conclusions?

2)Is it better to organize the summarized content through a clear logical relationship?

3)The manuscript lacks analysis and comparison of the differences and logical relations between different articles.

Author Response

Thank you for the useful comments. We have revised our manuscript and answered all the requirements.

1) The number of articles included in the systematic review may appear small, but the quality of information included in it, by a thoroughly and strictly obeyed inclusion criteria is important. This domain of research - metabolomics in gastric carcinogenesis is an extremely promising area, but still in its infancy, since there are scarce publications in the last decades on a restrained number of subjects.

2) We have reordered and corrected the pathogenic pathway of metastasis in a logical timeframe of events, from the immune system avoidance, cell cycle check-points elusion to invasion and distant tumour formation, both in Abstract and Results section.

3) Additional information has been provided in the Materials and Methods section and also in the Discussion section regarding all the selection process, study limitations and also in between studies comparison, in order to better emphasize the relationship between the studies.